# An Efficient Matrix Multiplication Algorithm for Accelerating Inference in Binary and Ternary Neural Networks

Mohsen Dehghankar [1]  Mahdi Erfanian [1]  Abolfazl Asudeh [1]

## Abstract

Despite their tremendous success and versatility, Deep Neural Networks (DNNs) such as Large Language Models (LLMs) suffer from inference inefficiency and rely on advanced computational infrastructure. To address these challenges and make these models more accessible and cost-effective, in this paper, we propose algorithms to improve the inference time and memory efficiency of DNNs with binary and ternary weight matrices. Particularly focusing on matrix multiplication as the bottleneck operation of inference, we observe that, once trained, the weight matrices of a model no longer change. This allows us to preprocess these matrices and create indices that help reduce the storage requirements by a logarithmic factor while enabling our efficient inference algorithms. Specifically, for a $n \times n$ weight matrix, our efficient algorithm guarantees a time complexity of $O(\frac{n^2}{\log n})$, a logarithmic factor improvement over the standard vector-matrix multiplication. Besides theoretical analysis, we conduct extensive experiments to evaluate the practical efficiency of our algorithms. Our results confirm the superiority of our approach both with respect to time and memory, as we observed a reduction in the multiplication time up to 29x and memory usage up to 6x. When applied to LLMs, our experiments show up to a 5.24x speedup in the inference time.

## 1. Introduction

Deep Neural Networks (DNNs) such as Large Language Models (LLMs) have achieved remarkable success and demonstrated versatility across a wide range of domains, yet they encounter significant challenges related to inference in-efficiency. These models demand substantial computational resources, including specialized, costly GPUs with ample memory to achieve real-time inference. This inefficiency leads to slower response times, elevated energy consumption, higher operational expenses, and, particularly, limited accessibility for everyday users who lack access to advanced computational infrastructure. For instance, current deployments of LLMs on typical consumer devices rely predominantly on API calls to powerful, remote servers (OpenAI, 2024; AI21 Labs, 2024; Hu et al., 2021; Hugging Face, 2023). While this approach enables users to leverage LLMs without needing advanced hardware, it introduces additional costs and delays due to network dependency, along with potential privacy concerns stemming from data transmitted to and processed by external servers (Yao et al., 2024; Pearce et al., 2023; Das et al., 2024; Finlayson et al., 2024).

Consequently, optimizing inference time and memory efficiency on standard, widely available hardware has become essential to make DNNs more practical and accessible for broader, real-world applications.

To that end, recent efforts have focused on quantizing the weights of DNNs, to enhance their computational efficiency and reduce energy consumption (Moons et al., 2017; Hubara et al., 2018; Wang et al., 2023; Chen et al., 2024; Ma et al., 2024). For example, limiting the weights to ternary values $\{-1, 0, 1\}$ in 1.58-bit LLMs has demonstrated to preserve accuracy comparable to that of general LLMs, thereby offering a more effective alternative for inference tasks (Ma et al., 2024).[1]

Expanding on recent advancements of DNNs with binary and ternary weights, in this paper, we propose algorithms that improve their inference time and memory efficiency. Our approach makes deploying these models viable on a broader range of devices, including those with limited computational power, ultimately making these tools more widely accessible and cost-effective.

Specifically, while viewing the inference process as sequences of multiplying activation vectors to weight matrices, we make a critical observation: *once the model is trained, the weight matrices remain fixed and do not change.*

---

[1] Department of Computer Science, University of Illinois Chicago, Chicago, USA. Correspondence to: Mohsen Dehghankar <mdehgh2@uic.edu>.

*Proceedings of the $42^{nd}$ International Conference on Machine Learning*, Vancouver, Canada. PMLR 267, 2025. Copyright 2025 by the author(s).

---

[1] Related Work is further discussed in Appendix B.

Following this observation, while focusing on matrix multiplication as the bottleneck operation, we preprocess the weight matrices of the trained model and create indices that enable efficient multiplication during the inference time. At a high level, our indices are bucketized permutation lists. Interestingly, by replacing each weight matrix with its preprocessed indices, our approach *reduces the space complexity* of storing the weights *by a logarithmic factor*.

We propose two algorithms for efficient multiplication of input vectors to the preprocessed weight matrices. When the weight matrices are ternary, our algorithms first transform those into pairs of binary matrices. Next, each binary matrix is partitioned into a set of column blocks. Then, permuting the rows based on a lexical order, we compute a set of aggregate values that contribute to different elements of the result vector. This process guarantees a time complexity of $O(\frac{n^2}{\log(n) - \log\log(n)})$ for $n \times n$ matrices in our first algorithm. Furthermore, introducing an efficient approach for writing the aggregate values in the result vector, our second algorithm improves the time complexity to $O(\frac{n^2}{\log(n)})$. The run-time of our algorithms further improves by fine-tuning their blocking parameter. Many widely used advanced DNNs are characterized by weight matrices of substantial sizes. For instance, the matrix size of GPT-3 is 12,288 ($\approx 2^{13}$) (Tsai, 2020; Tech, 2023; Brown, 2020), and this value is even greater for GPT-4. Consequently, achieving even a logarithmic factor improvement can have a significant impact.

In addition to theoretical analysis, we perform rigorous experiments to evaluate the time and memory of our algorithms in practice. Confirming our theoretical findings, our experiments demonstrate the superiority of algorithms over the standard $O(n^2)$ multiplications approach. In particular, **Our algorithms achieved up to a 29x reduction in inference time and a 6x reduction in memory usage for vector-matrix multiplication. When applied to 1.58-bit LLMs, the inference speedup is up to 5.24x.**

### 1.1. Paper Organization and Summary of Contribution

- Section 2: We formalize the vector-ternary-matrix multiplication problem and demonstrate its reduction to the vector-binary-matrix multiplication problem.

- Section 3: Given the weight matrices of a trained model, we preprocess them and construct indices that enable the development of our efficient algorithms while reducing the memory requirements by a logarithmic factor.

- Section 4: We introduce the RSR algorithm with a time complexity of $O\left(\frac{n^2}{\log(n) - \log(\log(n))}\right)$ for vector-binary-matrix multiplication for $n$ by $n$ matrices. We further optimize this algorithm and introduce RSR++ that achieves a faster running time of $O(\frac{n^2}{\log(n)})$.

We explain the generalization of our algorithms beyond binary and ternary settings in Section 4.4 and discuss parallelization in Appendix C.1.

- Sections 5: We conduct various experiments to demonstrate the applicability of our algorithms for matrix multiplication using different implementation configurations. We show that we can achieve up to 29x faster run time and 6x less space usage on matrix multiplication. We also conduct experiments on 1.58-bit LLMs. We discuss our approach's advantages and limitations in Appendix D.

## 2. Preliminaries

We consider the quantized DNNs with binary and ternary weights. The computation bottleneck of the inference process is a sequence of activation vector to weight matrix multiplications – the primary focus of our work. [2] In this section, we formally build the necessary notations, followed by our problem formulation. To simplify the explanations, we use ternary weights as a generalization of the binary weights. As a result, all of our algorithms are readily applicable to DNNs with either binary or ternary weights.

### 2.1. Notation

We denote vectors by $\vec{v}$ and use capital letters to refer to matrices. For example, $A \in E^{n \times m}$ is a matrix of size $n \times m$ with elements belonging to a set of permissible values, $E$. Specifically, if $E = \{0, 1\}$ (resp. $E = \{-1, 0, 1\}$), then $A \in E^{n \times m}$ is denoted as a **binary** (resp. **ternary**) matrix.

Let $\Sigma_n$ denote the set of all bijective permutation functions $\sigma : \{1, 2, \ldots, n\} \to \{1, 2, \ldots, n\}$. For a vector $\vec{v}$, the permuted vector under $\sigma$ is represented as $\pi_\sigma(\vec{v})$, where each element is repositioned such that $\pi_\sigma(\vec{v})[i] = \vec{v}[\sigma(i)]$, moving the element $\sigma(i)$ in $\vec{v}$ to position $i$ of the permuted vector. We extend $\pi_\sigma$ to matrices by permuting their rows. When $\sigma$ is clear by the context, we may simplify $\pi_\sigma$ to $\pi$. Let $\mathcal{L}_n^{\mathbb{N}}$ denote the set of all ordered lists of length $n$ with entries in $\mathbb{N}$, and let $\mathcal{L}_{<n}^{\mathbb{N}}$ represent the set of sorted lists with length at most $n$. We use $A[:, j]$ to indicate the $j$-th column of a matrix $A$. Similarly, we use $A[i, :]$ to refer to the $i$-th row the matrix $A$ and $A[i, j]$ to indicate an element.

### 2.2. Problem Formulation

Given a vector $\vec{v} \in \mathbb{R}^n$ and a ternary matrix $A \in \{-1, 0, 1\}^{n \times n}$, our objective is to efficiently compute the product $(\vec{v} \cdot A)$. [3] In this setting, the vector $\vec{v}$ serves as the activation output from the previous layer of the neural network, and $A$ is the weight matrix (See Figures 7 and 8 in

---

[2] Please refer to Appendix A for a background review.

[3] While all our algorithms and definitions are designed for any $n \times m$ matrix, to simplify our complexity analysis, we assume $A$ is a square matrix.

Appendix A). Note that, while the vector $\vec{v}$ is provided at inference time, weight matrices are fixed during the inference time.

Here, we focus on a single multiplication instance and aim to accelerate this operation beyond the quadratic time complexity of the standard vector-matrix multiplication.

**Problem 1 (Vector-Ternary-Matrix Product).** Given an input vector $\vec{v} \in \mathbb{R}^n$ and a pre-defined ternary matrix $A \in \{-1, 0, 1\}^{n \times n}$, compute the product $\vec{v} \cdot A$.

### 2.3. Solution Overview

Initially, we use the following proposition to reduce our problem into a Vector-Binary-Matrix product.

**Proposition 2.1.** *Any ternary matrix $A$ can be expressed as $A = B^{(1)} - B^{(2)}$, where $B^{(1)}$ and $B^{(2)}$ are the following binary matrices:*

$$B^{(1)}[i,j] = \begin{cases} 0 & \text{if } A[i,j] \in \{-1, 0\}, \\ 1 & \text{if } A[i,j] = 1 \end{cases} \tag{1}$$

$$B^{(2)}[i,j] = \begin{cases} 0 & \text{if } A[i,j] \in \{0, 1\}, \\ 1 & \text{if } A[i,j] = -1 \end{cases}$$

Therefore, we focus on solving the following problem.

**Problem 2 (Vector-Binary-Matrix Product).** Given an input vector $\vec{v} \in \mathbb{R}^n$ and a pre-defined binary matrix $B \in \{0, 1\}^{n \times n}$, compute the product $\vec{v} \cdot B$.

Following Proposition 2.1, any guarantees established for Problem 2 equivalently apply to Problem 1 by a constant factor. We introduce two algorithms to efficiently solves Problem 2 (hence Problem 1): (1) **R**edundant **S**egment **R**eduction (RSR) and (2) RSR++.

Figure 1 provides an overview of our approach. Our algorithms identify and reuse redundant computations, by first preprocessing the weight matrices and constructing essential data structures utilized during inference time. The preprocessing phase is explained in Section 3. Subsequently, in Section 4, we describe the inference-time operations, illustrating how our algorithms accelerate the multiplication process, reducing the quadratic complexity of the standard vector-matrix-multiplication by a logarithmic factor.

## 3. Preprocessing: Index Construction

In this section, we outline the preprocessing phase of our approach, during which we construct the essential data structures for the matrix $B$ in Problem 2. These structures are designed to optimize and accelerate the multiplication process during the inference time (explained in Section 4).

Given a ternary matrix $A = W^i$, representing the weights between two layers of a ternary NN, we first present it as the subtraction of two binary matrices $A = B^{(1)} - B^{(2)}$, as described in Proposition 2.1. Let $B$ be either $B^{(1)}$ or $B^{(2)}$. At a high level, during the preprocessing time, we partition the columns of $B$ into a set of *column blocks*, associating each block a row-permutation as its index to identify the longest common segments across the columns. This permutation is used later at the inference time for efficient vector-to-matrix multiplication.

The space complexity for representing a matrix $B$ is $O(n^2)$. Interestingly, as we shall explain in Section 3.4.1, the space complexity of our indices is $O(\frac{n^2}{\log n})$, *reducing a logarithmic factor in the space* required for representing $B$.

### 3.1. Step 1: Column Blocking

The first step of our preprocessing is *Column Blocking* – the process of partitioning consecutive columns of the original binary matrix $B$ to construct a set of smaller, compact matrices, each representing a distinct block of columns.

**Definition 3.1 ($k$-Column Block).** Let $B \in \mathbb{R}^{n \times n}$ and $i \in \{1, 2, \cdots, \lceil \frac{n}{k} \rceil\}$. We define $B_i^{[k]}$ as an $n \times k$ matrix containing columns $(i-1) \cdot k + 1$ to through $\min(i \cdot k, n)$. This construction yields a series of submatrices $B_i^{[k]}$ that partition $B$ into $\lceil \frac{n}{k} \rceil$ contiguous column blocks. Each $B_i^{[k]}$ is called a $k$-Column Block of $B$.[4]

For example, consider the following binary matrix,

$$B = \begin{bmatrix} 0 & 1 & 1 & 1 & 0 & 1 \\ 0 & 0 & 0 & 1 & 1 & 1 \\ 0 & 1 & 1 & 1 & 1 & 0 \\ 1 & 1 & 0 & 0 & 1 & 0 \\ 0 & 0 & 1 & 1 & 0 & 1 \\ 0 & 0 & 0 & 0 & 1 & 0 \end{bmatrix}$$

The set of 2-Column Blocks of $B$ are as follows:

$$B_1^{[2]} = \begin{bmatrix} 0 & 1 \\ 0 & 0 \\ 0 & 1 \\ 1 & 1 \\ 0 & 0 \\ 0 & 0 \end{bmatrix}, B_2^{[2]} = \begin{bmatrix} 1 & 1 \\ 0 & 1 \\ 1 & 1 \\ 0 & 1 \\ 1 & 0 \\ 0 & 1 \end{bmatrix}, B_3^{[2]} = \begin{bmatrix} 0 & 1 \\ 1 & 1 \\ 1 & 0 \\ 1 & 0 \\ 0 & 1 \\ 1 & 0 \end{bmatrix}$$

When $k$ is clear by the context, we use $B_i$ instead of $B_i^{[k]}$ to denote the $i$-th block.

---

[4]This definition can be extended to any $B \in \mathbb{R}^{m \times n}$.

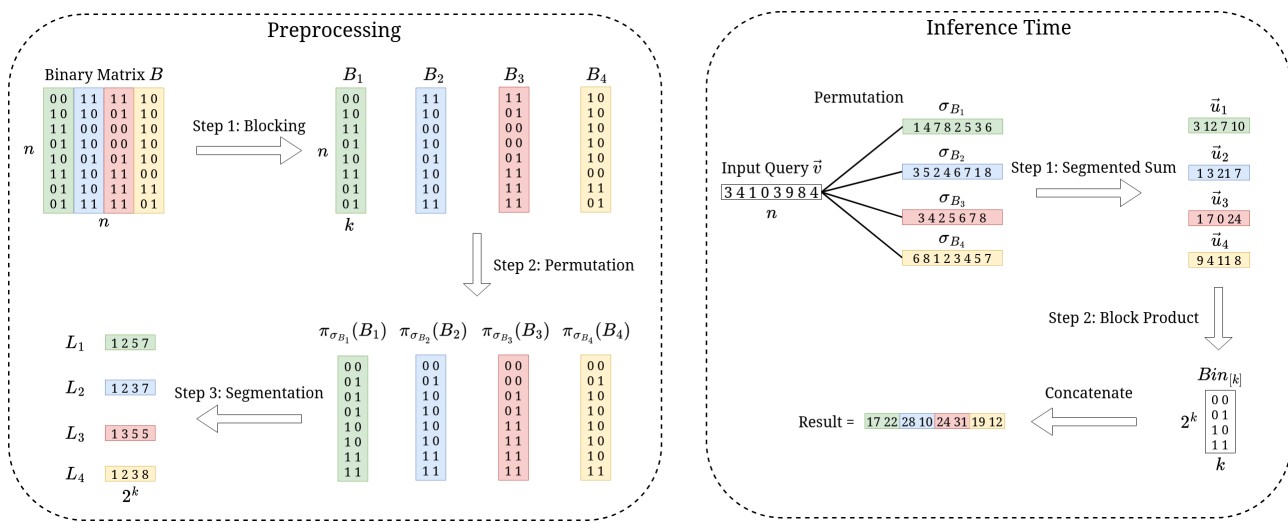

Figure 1: A visualization of the Redundant Segment Reduction method. The calculation of $\vec{v} \cdot B$. In this example, $k = 2$.

### 3.2. Step 2: Row Permutation

The next step after column blocking is *binary row ordering*, where the rows of each column block are sorted according to a lexicographic order.

**Definition 3.2** (Binary Row Order). Let $B_i \in \mathbb{R}^{n \times k}$ be a binary matrix. For each row $B_i[r, :]$, let $B_i[r, :]_2$ be the corresponding binary value of concatenating $B_i[r, 1] \cdots B_i[r, k]$. For example, if $B_i[r, :] = [1, 0, 1, 1]$, then $B_i[r, :]_2 = (1011)_2$. The Binary Row Order of $B_i$, denoted as $\pi_{\sigma_{B_i}}$, is defined as a permutation on $B_i$, such that the rows of $\pi_{\sigma_{B_i}}(B_i)$ are sorted based on their corresponding binary value in ascending order. That is, $\forall r \neq s \leq n$, if $B_i[r, :]_2 < B_i[s, :]_2$, then $\sigma_{B_i}(r) < \sigma_{B_i}(s)$. We call $\pi_{\sigma_{B_i}}(B_i)$ as the Binary Row Order of $B_i$.[5]

To further clarify the Binary Row Order, let us consider Example 3.3.

*Example* 3.3. Let $B_i$ be a block with two columns. The permutation function $\sigma_{B_i} = \langle 2, 5, 6, 1, 3, 4 \rangle$[6] provides a Binary Row Order $\pi_{\sigma_{B_i}}(B_i)$ of the matrix $B_i$.

$$
B_i = \begin{bmatrix} 0 & 1 \\ 0 & 0 \\ 0 & 1 \\ 1 & 1 \\ 0 & 0 \\ 0 & 0 \end{bmatrix} \implies \pi_{\sigma_{B_i}}(B_i) = \begin{bmatrix} 0 & 0 \\ 0 & 0 \\ 0 & 0 \\ 0 & 1 \\ 0 & 1 \\ 1 & 1 \end{bmatrix} \quad (2)
$$

The right expression results from $00_2 < 01_2 < 11_2$. □

---

[5]In general, any binary matrix following this lexicographic order is called Binary Row Ordered matrix in this paper.

[6]$\sigma_{B_i}(j)$ is the $j$-th value in $\sigma_{B_i}$. For example, $\sigma_{B_i}(2) = 5$.

We define $Bin_{[k]}$ as a binary $2^k \times k$ matrix which has exactly one row for each binary value from $0$ to $2^k - 1$, and it is also Binary Row Ordered. For example,

$$
Bin_{[1]} = \begin{bmatrix} 0 \\ 1 \end{bmatrix}, Bin_{[2]} = \begin{bmatrix} 0 & 0 \\ 0 & 1 \\ 1 & 0 \\ 1 & 1 \end{bmatrix}, Bin_{[3]} = \begin{bmatrix} 0 & 0 & 0 \\ 0 & 0 & 1 \\ 0 & 1 & 0 \\ 0 & 1 & 1 \\ 1 & 0 & 0 \\ 1 & 0 & 1 \\ 1 & 1 & 0 \\ 1 & 1 & 1 \end{bmatrix}
$$

### 3.3. Step 3: Segmentation

Next, we *segment* each binary row ordered matrix (the sorted column blocks) into groups of rows with the same binary value representation, enabling further matrix compaction.

**Definition 3.4** (Segmentation List). Given a Binary Row Ordered matrix $B$ of size $n \times k$, the function $\mathcal{S}(B)$ returns the list of the boundary indices, specifying the ranges of rows with the same binary value representations.

For example, the segmentation on a column block $B_i$ is defined as $\mathcal{S}(\pi_{\sigma_{B_i}}(B_i))$. For simplicity, we denote this segmentation as $\mathcal{S}(B_i)$. Specifically, let $\ell = \mathcal{S}(B_i)[j]$. Then, all rows in range $[\ell, \mathcal{S}(B_i)[j + 1])$, have the binary value representation $B_i[\ell, :]_2$. Note that $\min\{2^k, n\}$ provides an upper bound on the size of $\mathcal{S}(B_i)$.

Consider the Binary Row Ordered matrix $\pi_{\sigma_{B_i}}(B_i)$ shown in Example 3.3. The Segmentation list of this matrix is $[1, 4, 6]$ because the first row is the beginning of $00$ rows. Then, $01$ starts from index $4$, and $11$ starts at index $6$.

| Row Binary Value | 00 | 01 | 10 | 11 |
|---|---|---|---|---|
| Start Index | 1 | 4 | 6 | 6 |

Figure 2: The Full Segmentation of Example 3.3. There is no starting index for row 10, so we skip it by using the same start index of next available value. The Full Segmentation list is the second row of the table.

---

**Algorithm 1** `Preprocess`

---

1: **Input:** binary matrix $B$, and a parameter $k$
2: **Output:** permutations $\sigma_{B_i}$ and segmentations $L_i = \mathcal{S}(B_i)$
3: Create $k$-Column blocks $B_i$          ▷ *Blocking*
4: **for** each block $B_i$ **do**
5:     $\sigma_{B_i} \leftarrow$ Binary Row Order of $B_i$    ▷ *Permutation*
6:     $L_i \leftarrow$ Segments of $\pi_{\sigma_{B_i}}(B_i)$    ▷ *Segmentation*
7: **end for**
8: **Return:** $\{\sigma_{B_i} \mid \forall i\}$ and $\{L_i \mid \forall i\}$

---

For a Binary Row Ordered matrix of dimensions $n \times k$, we extend this concept to define **Full Segmentation** as the list of exactly $2^k$ elements, where the $j$th element of the list indicates the first index of a row that has the binary value $j$. For example, the Full Segmentation of matrix $\pi_{\sigma_{B_i}}(B_i)$ in Example 3.3 is $[1, 4, 6, 6]$. Since there are no rows for 10, the same boundary is assigned for the next value. See Figure 2 for an illustration.

From now on, we use the same notation $\mathcal{S}$ to denote the Full Segmentation for a Binary Row Ordered matrix.

**Proposition 3.5.** *For a binary matrix $B_i \in \{0,1\}^{n \times k}$ and $j < 2^k$, $\mathcal{S}(B_i)[j+1] - \mathcal{S}(B_i)[j]$ represents the number of rows in $B_i$ whose binary value corresponds to $j$. The number of rows for $j = 2^k$ is $n + 1 - \mathcal{S}(B_i)[j]$.*

Given the Full Segmentation of a Binary Row Ordered matrix, a compact representation of the matrix can be obtained by retaining only the first indices of each unique row value in the segmentation list. In the following sections, for simplicity, we use $L_i$ instead of $\mathcal{S}(B_i)$ to show the Full Segmentation of the permuted Column Block $B_i$.

### 3.4. Preprocessing Algorithm

Putting all three steps together, the preprocessing algorithm is described in Algorithm 1. The algorithm starts by selecting a parameter $k$ such that $k \leq \log_2(n)$. Next, it computes the $k$-Column Blocks of $B$. Hereafter, we denote these blocks as $B_i$ rather than $B_i^{[k]}$ for $i \leq \lceil \frac{n}{k} \rceil$. For each $B_i$, the algorithm proceeds to calculate both the Binary Row Orders $\sigma_{B_i}$ and the Full Segmentations $L_i$. Figure 1 (Preprocessing – the left figure) provides a visual example of the preprocessing steps.

3.4.1. COMPLEXITY ANALYSIS

**Theorem 3.6.** *Representing a binary or ternary weight matrix as block indices requires a space complexity $O\left(\frac{n^2}{\log(n)}\right)$, and the preprocessing algorithm to generate the block indices has a time complexity of $O(n^2)$.*

## 4. Inference Time: Vector-to-Matrix Multiplication

To simplify our analysis, without loss of generality, let us assume the weight matrices are of size $n \times n$. Specifically, given a preprocessed $n \times n$ binary matrix $B$, we aim to efficiently compute the product $\vec{v} \cdot B$ when a vector $\vec{v} \in \mathbb{R}^n$ is provided at *inference time* (see Figure 8 in Appendix A ).

### 4.1. Segmented Sum Computation

Recall that during the preprocessing time, for each column block $B_i$, we construct a permutation function $\sigma_{B_i}$ and a full segmentation list $L_i$ as its index.

At the inference time, given a vector $\vec{v}$, our objective is to *efficiently* compute the multiplication of $\vec{v}$ by each column block $B_i$, using $(\sigma_{B_i}, L_i)$.

Our first step is computing the **Segmented Sums** over the permuted vector $\pi_{\sigma_{B_i}}(\vec{v})$ – i.e., the sum for each interval in the permuted vector that corresponds to groups of similar rows in the binary matrix $B_i$.

**Definition 4.1** (Segmented Sum). Consider the permutation function $\sigma_{B_i}$ and the full segmentation list $L_i$, for a column block $B_i$. Let $\vec{v}_\pi = \pi_{\sigma_{B_i}}(\vec{v})$ be the permutation of the vector $\vec{v} \in \mathbb{R}^n$. The Segmented Sum of $\vec{v}_\pi$ on $L_i$ is defined as a vector $SS_{L_i}(\vec{v}_\pi)$ of size $|L_i|$ where,

$$SS_{L_i}(\vec{v}_\pi)[j] = \begin{cases} \sum_{k=L[j]}^{L_i[j+1]-1} \vec{v}_\pi[k] & \text{if } j < |L_i| \\ \\ \sum_{k=L_i[j]}^{n} \vec{v}_\pi[k] & \text{if } j = |L_i| \end{cases} \quad (3)$$

For example, for the Full Segmentation defined for Example 3.3 and a given vector $\vec{v} = [3, 2, 4, 5, 9, 1]$, we have:

$$SS(\vec{v}) = [9, 14, 0, 1] \quad (4)$$

Where $9 = 3 + 2 + 4, 14 = 5 + 9, 1 = 1$, and the third element is 0 because there is no $10_2$ in the segmentation.

Computing the Segmented Sums using Equation 3 requires first computing the permuted vector $\vec{v}_\pi$. Instead, to efficiently compute the Segmented Sums in place (without

---

**Algorithm 2** RSR (Inference Time)

---

1: **Input:** vector $\vec{v} \in \mathbb{R}^n$, binary matrix $B$
2: **Output:** result $\vec{r} = \vec{v} \cdot B$ where $\vec{r} \in \mathbb{R}^n$
3: **for** each block $B_i$ **do**
4:     $\vec{u} \leftarrow SS_{L_i, \sigma_{B_i}}(\vec{v})$     ▷ *Step 1; Segmented Sum (Eq 5)*
5:     $\vec{r}_i \leftarrow \vec{u} \cdot Bin_{[k]}$     ▷ *Step 2; Block Product*
6: **end for**
7: $\vec{r} \leftarrow (\vec{r}_1, \vec{r}_2, \cdots, \vec{r}_{\lceil \frac{n}{k} \rceil})$     ▷ *Concatenate block results*
8: **Return:** $\vec{r}$

---

computing the permuted vector), we use Equation 5:

$$SS_{L_i}(\vec{v}_\pi)[j] = \qquad\qquad (5)$$

$$SS_{L_i, \sigma_{B_i}}(\vec{v})[j] = \begin{cases} \sum_{k=L_i[j]}^{L_i[j+1]-1} \vec{v}[\sigma_{B_i}(k)] & \text{if } j < |L_i|, \\ \\ \sum_{k=L_i[j]}^{n} \vec{v}[\sigma_{B_i}(k)] & \text{if } j = |L_i| \end{cases}$$

### 4.2. RSR

We are now ready to describe our algorithm RSR for computing $\vec{v}.B$ at the inference time. Given the input vector $\vec{v}$, for each column block $B_i$, in **Step 1** we use the permutation function $\sigma_{B_i}$ and the full segmentation list $L_i$ to compute the Segmented Sum $SS_{L_i, \sigma_{B_i}}(\vec{v})$ using Equation 5.

Then, we use Lemma 4.2 to calculate $\vec{v} \cdot B_i$.

**Lemma 4.2.** $\vec{v} \cdot B_i = SS_{L_i, \sigma_{B_i}}(\vec{v}) \cdot Bin_{[k]}$.

Following Lemma 4.2, as the final step in the inference time, in **Step 2** we calculate the product $SS_{L_i, \sigma_{B_i}}(\vec{v}) \cdot Bin_{[k]}$ for each $k$-Column Block $B_i$. Algorithm 2 shows a pseudo-code of this algorithm.

#### 4.2.1. COMPLEXITY ANALYSIS

For each column block $B_i$, step 1 takes $O(n)$ since it makes a pass over each element of $\vec{v}$ exactly once. Step 2 computes the product of a vector of size $2^k$ with a matrix of dimensions $2^k \times k$, resulting in a time complexity of $O(k \cdot 2^k)$.

The entire process is applied to $\frac{n}{k}$ $k$-Column Blocks. Consequently, the total query time is $O\left(\frac{n}{k}(n + k \cdot 2^k)\right)$. For any selection of $k \cdot 2^k \leq n$, the overall running time can be written as $O\left(\frac{n^2}{k}\right)$.

Specifically, setting $k = \log(\frac{n}{\log(n)})$ results in a time complexity of $O\left(\frac{n^2}{\log(\frac{n}{\log(n)})}\right) = O\left(\frac{n^2}{\log(n)-\log(\log(n))}\right)$.

**Theorem 4.3.** *The* RSR *algorithm solves Problem 1 with the time complexity of* $O\left(\frac{n^2}{\log(n)-\log(\log(n))}\right)$.

#### 4.2.2. THE OPTIMAL $k$

While we could prove Theorem 4.3 by choosing $k = \log(\frac{n}{\log(n)})$, this choice might not be optimal, i.e., it may

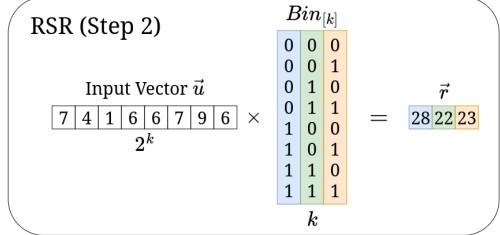

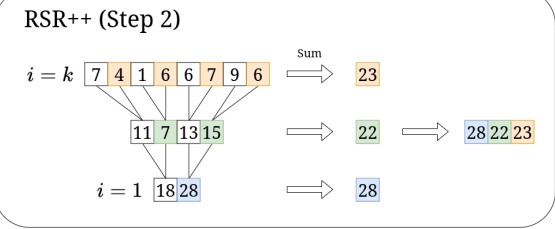

Figure 3: Visualizing Step 2 of RSR++ (Algorithm 3) versus RSR at inference time.

not minimize the run-time of the algorithm.

The optimal value of $k$ can be found using Equation 6.

$$k^{opt} = \underset{k \in [\log(\log(n))]}{\arg\min} \frac{n}{k}(n + k2^k) \qquad (6)$$

To find the optimal value of $k$ based on Equation 6, we apply a *binary search* on $k$ in the range $[1, \log(n) - \log(\log(n))]$. In Appendix F.1, we run experiments to find the optimal value of $k$ and to evaluate the impact of varying it.

### 4.3. RSR++

Having presented RSR, we now present our faster algorithm RSR++ that focuses on improving Step 2 of RSR. This step involves computing the product of a vector $\vec{u}$ of size $2^k$ and the binary matrix $Bin_{[k]}$ of size $2^k \times k$ (Line 2 of Algorithm 2). The standard vector-matrix multiplication, in this case, leads to $O(k \cdot 2^k)$ time. However, the special structure of matrix $Bin_{[k]}$ allows us to reduce it to $O(2^k)$. Algorithm 3 shows the pseudo-code for this multiplication. See Figure 3 for a visualization of this approach.

Let $\vec{r}$ denote the final result of this product, $\vec{u} \cdot Bin_{[k]}$. We build $\vec{r}$ starting from the $k$-th element to the first one. (i) The $k$-th element can be calculated by summing up the elements in $\vec{u}$ with odd indices. For example, in RSR++ visualization in Figure 3, the sum of odd-index elements (orange cells) in the top row is 23. Then, (ii) we calculate a vector $\vec{x}$ of size $\frac{|\vec{u}|}{2}$ where we sum each two consecutive elements of $\vec{u}$. Now we can see that the $(k-1)$-th element of $\vec{r}$ can be calculated by doing the same process as (i), this time, on $\vec{x}$ (see rows 2 and 3 in RSR++ visualization in Figure 3). We

continue this process by repeating steps (i) and (ii) to build all $k$ elements of $\vec{r}$.

### 4.3.1. COMPLEXITY ANALYSIS

Step (i) is linear in terms of the size of $\vec{x}$ at each step (see Line 5 of Algorithm 3). As a result, the running time is $\sum_{i=k}^{1} O(2^i) = O(2^k)$. Step (ii) also takes linear in size of $\vec{x}$. Consequently, the overall time calculating $\vec{u} \cdot Bin_{[k]}$ using this subroutine takes $O(2^k)$.

We can follow the same process of analyzing RSR, but this time using RSR++ as a subroutine for Line 5 of Algorithm 2. Based on the same analysis in Section 4.2.1, the inference time would be $O(\frac{n}{k}(n + 2^k))$, for a $k = \log(n)$ this would result in the running time of $O\left(\frac{n^2}{\log(n)}\right)$.

**Theorem 4.4.** *The* RSR++ *algorithm solves Problem 1 with a time complexity of* $O\left(\frac{n^2}{\log(n)}\right)$.

### 4.3.2. THE OPTIMAL $k$

The process for finding the optimal value of $k$ for RSR++ is the same as the one for RSR, except that the binary search is applied in the range $[1, \log(n)]$ to optimize the following equation:

$$k^{opt} = \underset{k \in [\log(n)]}{\operatorname{argmin}} \quad \frac{n}{k}(n + 2^k) \tag{7}$$

### 4.4. Generalization

Our algorithms naturally extend to vector multiplication with any $q$-bit matrix. Specifically, given a matrix $A$ where each entry is encoded using $q$ bits, we can decompose $A$ into a weighted sum of binary matrices:

$$A = \sum_{k=0}^{q-1} 2^k \cdot B_k,$$

where each $B_k$ is a binary matrix in which the element $B_k[i, j]$ is 1 if and only if the $k$th bit of $A[i, j]$ is set. We then apply the RSR algorithm separately to each $B_k$, and combine the resulting products. This extension incurs only a linear overhead in runtime with respect to $q$.

### 4.5. Parallelization

Due to the independence of computations across column blocks, our algorithms are naturally parallelizable. We discuss the parallelization of our algorithms on CPU and GPU in Appendix C.1.

## 5. Experiments

In this section, we evaluate our proposed algorithms' time and memory performance in real-world scenarios. Our codes are publicly available in this repository. In addition, some of our implementation details for CPU and GPU experiments are provided in Appendix E. In the following, first in Section 5.1, we detail the native C++ implementation of both RSR and RSR++, comparing their runtime performance under a fair setting to validate our theoretical guarantees. In Section 5.2, we implement and evaluate the algorithms using Python's NumPy package, providing insights into their performance in a more practical and widely used computational environment. Finally, in Sections 5.3 and 5.4 we conduct experiments on real 1.58-bit LLMs. Our extended experiments, including implementations in BitNet.cpp environment and per transformer module analysis, are provided in Appendix F. Specifically, Appendix F.1 explores the process of determining the optimal value of $k$, identifying the best $k$ for each input size $n$. A performance comparison between RSR and RSR++ is provided in Appendix F.2, followed by extended run-time analysis of RSR on NumPy in Appendix F.3. A performance evaluation of our algorithm using GPU is also provided in Appendix F.4.

### 5.1. Native Implementation

In this section, we focus on verifying the theoretical foundation of our proposed methods, RSR and RSR++, using native implementations in C++. We selected C++ as the implementation language because, being a compiled language, it is less susceptible to runtime noise, allowing inference time to better reflect the exact time complexity of the methods, relatively. As baselines, we also implemented Standard Matrix Multiplication, which we will refer to as Standard throughout this section.

For input, we assume access to a binary weight matrix $B^{n \times n}$ where $2^{11} \leq n \leq 2^{16}$ during preprocessing. During inference time, given input vector $\vec{v}$ of size $n$, our aim is to compute $v \cdot B$. with both $B$ and $\vec{v}$ values generated randomly. These parameters align with typical size of weight matrices and input vector patterns commonly encountered in deep learning and large language model (LLM) inference tasks. We then measure inference time across varying values of $n$ for each method. We also utilized the optimal value of $k$ for each $n$, which is discussed in detail in Appendix F.1. Figure 4 presents the inference times for these methods. Notably, **our proposed algorithms achieve up to a 29x speedup** over the Standard baseline when $n = 2^{16}$, representing a substantial improvement that could significantly impact DNN and LLM architectures, enabling them to utilize binary and ternary weights more effectively.

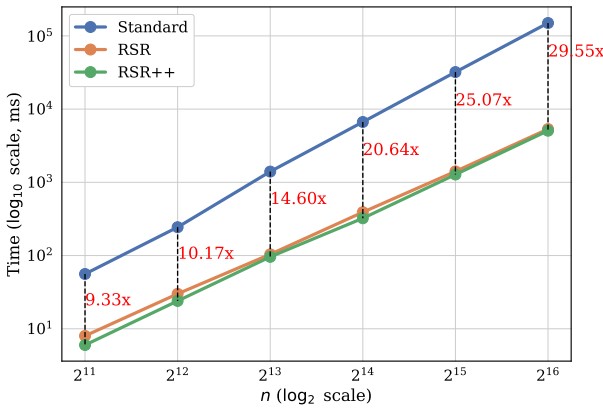

Figure 4: Comparison of `RSR`, `RSR++`, and Standard on native C++ implementation for Binary Matrix Multiplication. The speedup values are between `RSR++` and Standard. Each value is the average of 10 different runs.

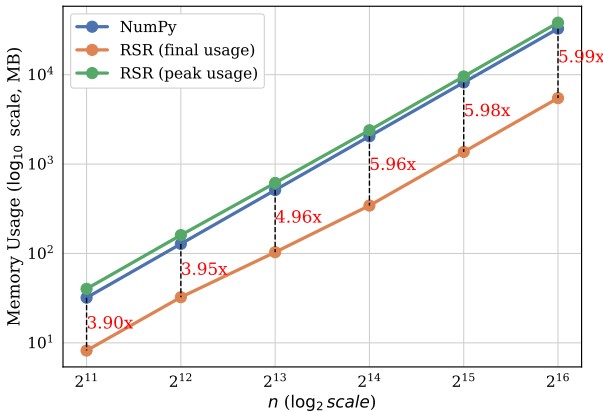

Figure 5: Memory consumption of `RSR` after the preprocessing is done. Compared to memory required for the Standard matrix multiplication (NumPy). In `RSR`, we only store permutations and segmentation lists in the memory.

### 5.2. Matrix Multiplication Using `NumPy`

In this section, we investigate the practical performance improvements achieved by `RSR`, extending beyond native simulation environments. To this end, we use `NumPy`, a state-of-the-art library for matrix multiplication upon which numerous high-performance computing libraries are based. We implement `RSR` in Python using `NumPy`'s built-in functionalities and compare the inference time and memory consumption between `RSR` and `NumPy`'s `np.dot()` function (referred to as *NumPy*). We provide our Inference time evaluation in Appendix F.3, where in summary, we observed an up to 24x faster inference times by our algorithm.

`RSR` requires a preprocessing step to compute permutations and segments, where only these two values need to be stored

to calculate the final weights. This preprocessing enables substantial compression. Figure 5 illustrates the memory consumption in megabytes for `NumPy` and `RSR`. In the peak of preprocessing procedure, both the weight matrix $A$ and the segment data are stored in memory (indicated by the green line); however, after computation, only the segmentation lists and permutations are stored and $A$ will be cleared from memory. For a matrix of size $n = 2^{16}$, the storage required is reduced to less than 17% of the original matrix size (**5.99x improvement**), highlighting the benefits of this approach for large-scale storage and data transfer.

In summary, `RSR` provides practical advantages for deploying large models. For instance, companies training new LLMs could preprocess their weights to release only the final segments, permutations, and the optimal parameter $k$. This would result in **up to 24x faster inference times and up to 5.99x memory reduction**, significantly easing both storage and transfer demands.

### 5.3. LLM Inference on CPU

To further evaluate the practical performance of our algorithms, we examine their application to quantized large language models by replacing the matrix multiplications in the fully connected layers. Specifically, we run experiments using the 1.58-bit quantized versions of **Llama3** and **Falcon3**.[7] The experiments were conducted on a server with the following specifications: a 16-core Intel Xeon CPU, an NVIDIA Tesla T4 GPU, 32 GB of RAM, and Debian 11 as the operating system. Our algorithms were implemented using the `PyTorch`[8] library in Python. The details of the implementation is discussed in the Appendix E.

In this experiment, we perform LLM inference on CPU to assess the speedup gain by our algorithms on a simple setting without parallalization. This setup simulates environments with limited computational resources, such as personal devices. Subsequently, in the Section 5.4, we shall extend our implementation to GPU, using the parallelization techniques outlined in Section C.1.

We utilized the 1.58-bit model implemented following the work in (Ma et al., 2024). Having downloaded the weights of the pre-trained model, we applied the pre-processing step of our algorithm on the weight matrices (this step is done only once per model). During the inference time, for each fully connected layer (`torch.nn.BitLinear`), we integrated and executed the inference step of `RSR`.

To evaluate the performance, we conducted experiments on three datasets. First, we generated a synthetic dataset of short factual questions (**ShortQuestions**) with assistance

---

[7]Link to Llama3-8B-1.58bit, Falcon3-3B-1.58bit , and Falcon3-10B-1.58bit on Huggingface.

[8]https://pytorch.org/

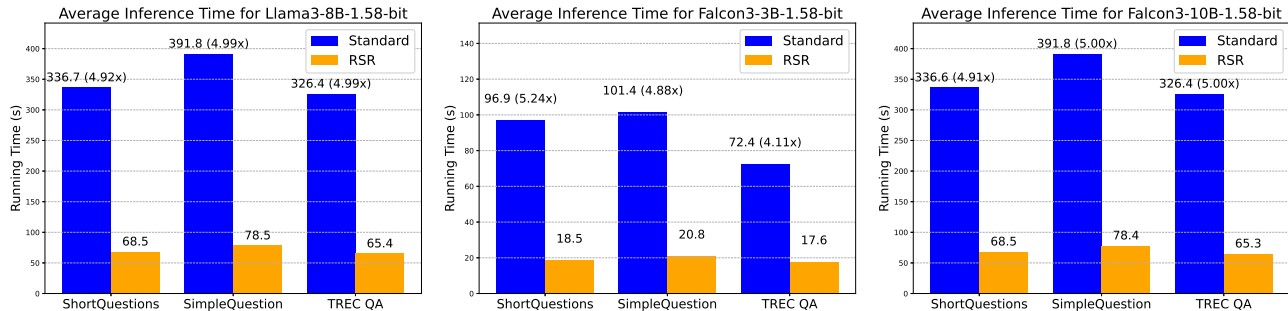

Figure 6: Average inference time on CPU for different 1.58-bit models. The x-axis represents the three different datasets.

from GPT-4[9]. The other two datasets used were **Simple-Questions** (Diefenbach et al., 2017) and **TREC QA** (Wang et al., 2007). The matrix sizes in the Llama3 model ranged from $2^{12}$ to $2^{13}$, while for Falcon3 models, they ranged from $2^{11}$ to $2^{12}$. For each input, we generated a single token by running one feedforward pass through the LLM and verified the equality of responses with and without applying `RSR`.

The average inference times are presented in Figure 6. `Standard` is the baseline 1.58-bit model without any change and `RSR` is our algorithm. As shown, the inference time of the LLM improves by up to **5.24x**. While the standard method benefits from low-level PyTorch optimizations for single vector-to-matrix multiplications[10], our algorithm operates solely at the application level without leveraging these optimizations. This introduces additional overhead, which reduces the observed speedup compared to experiments focused purely on matrix multiplication. However, the space usage of this implementation remains identical to the baseline, as the size of the segmentation matrix $\mathcal{M}$ is the same as the ternary weight matrix (See Appendix C.1 for the definition of segmentation matrix).

**5.4. LLM Inference on GPU**

Using the implementation detailed in Appendix E.3, we evaluated the inference time of the models with and without the `RSR` implementation on GPU, and the results are presented in Table 1. Notably, we achieved nearly a **2.5x** speedup, despite our algorithm being implemented at the application level, compared to the optimized matrix multiplication of PyTorch. Our experiments on naive vector-matrix multiplication on GPU is provided in Appendix F.4.

---

[9]See Appendix E.1 for more details.

[10]PyTorch default settings such as caching mechanisms, backend-specific optimizations (e.g., cuBLAS, cuDNN), and kernel fusion, are fully enabled.

| Model | Standard ($\mu$s) | RSR ($\mu$s) |
|---|---|---|
| Llama3-8B-1.58bit | $392 \pm 20$ | $\mathbf{225 \pm 29}$ |
| Falcon3-3B-1.58bit | $560 \pm 24$ | $\mathbf{206 \pm 21}$ |
| Falcon3-10B-1.58bit | $364 \pm 82$ | $\mathbf{210 \pm 24}$ |

Table 1: Average inference time on **GPU**. All values are in microseconds $\pm$ standard deviation.

## 6. Conclusion

This paper presented algorithms that significantly improve inference time and memory efficiency for quantized neural networks (like LLMs) with binary and ternary weights. By preprocessing weight matrices to create indices, our approach reduces storage complexity for maintaining the model weights by a logarithmic factor. It enhances inference efficiency, achieving a time complexity of $O(\frac{n^2}{\log n})$.

Furthermore, our experiment results across various settings demonstrated the practical advantages of our methods, with observed reductions in inference time of up to 29x and memory usage of up to 6x, underscoring the potential of our approach to make LLMs more accessible and cost-effective.

## Acknowledgments

This work was supported in part by the National Science Foundation, Grant No. 2348919 and 2107290. The authors would like to thank the anonymous reviewers for their time and invaluable feedback.

## Impact Statement

This paper presents work whose goal is to advance the field of Machine Learning. There are many potential societal consequences of our work, none which we feel must be specifically highlighted here.

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

# APPENDIX

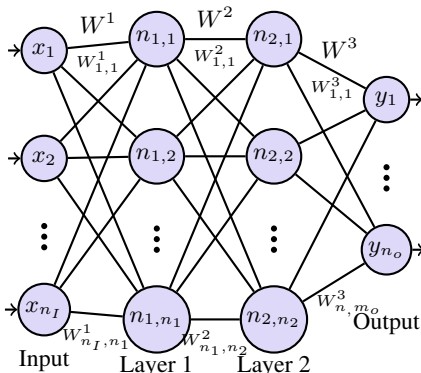

Figure 7: A feedforward neural network with $\ell = 3$ layers and $n_i$ nodes per hidden layer.

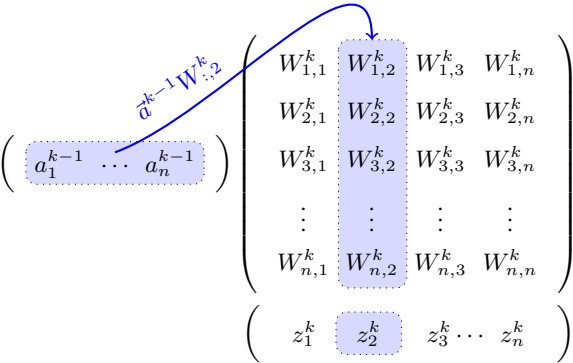

Figure 8: Illustration of vector-to-matrix multiplication as the core inference operation.

## A. Background

A trained DNN model consists of a collection of weights used for inference (Figure 7). The weights are learned during the training phase and are *fixed during the inference time*. During the inference time, given an input vector, the feedforward process can be viewed as a series of matrix multiplications applied to the input vector to produce the final output. For instance, consider the two consecutive layers shown in a multi-layer perceptron in Figure 7. Each weight matrix $W^i$ represents the edge weights between layers $(i - 1)$ and $i$ with layer 0 being the input vector. The feedforward step is a chain of matrix products and activation functions that maps input vector $\vec{x}$ to output vector $\vec{y}$; it can be represented as follows (with $\vec{a}^0 = \vec{x}$), where $f$ is the activation function (e.g., sigmoid or ReLU).

$$\vec{z}^k = \vec{a}^{k-1} W^k \qquad \vec{a}^k = f(\vec{z}^k)$$

Figure 8 visualizes the bottleneck operation of the inference process: the multiplication of each activation vector $\vec{a}^{k-1}$ to the weight matrix $W^k$ to generate the vector $\vec{z}^k$.

Building on this background, *our work focuses on accelerating the bottleneck operation, the vector-ternary-matrix product*. By improving the speed of this operation, we proportionally accelerate the entire *inference process* of a quantized LLM (or, more generally, a neural network), as each matrix product in the feedforward procedure is computed sequentially until the output layer. We reformulate this problem as a vector-binary-matrix multiplication, thereby enabling compatibility with both binary and ternary matrices.

## B. Related Work

In this paper, we consider the quantized DNNs with binary and ternary weights (Dai et al., 2021; Wu et al., 2016; McKinstry et al., 2018; Zhu et al., 2016; Courbariaux et al., 2015; Hubara et al., 2016).

Examples of ternary DNNs include (Moons et al., 2017; Hubara et al., 2018; Yuan et al., 2024) and 1.58-bit LLMs (Ma et al., 2024), where the weights are limited to $\{-1, 0, 1\}$.

Previous research has explored enhancing matrix multiplication through hardware-level accelerations (Guo et al., 2024; Wang et al., 2024b; Park & Choi, 2019; Trusov et al., 2022; Choi et al., 2021). In contrast, our approach focuses on designing an algorithm that provides a guaranteed logarithmic improvement over naive matrix multiplication at the application layer, independent of specific hardware architectures.

The Four Russians Algorithm (Arlazarov et al., 1970) is a classical method for efficient Boolean matrix multiplication. While the Four Russians Algorithm relies on tiling the matrix and computing the smaller products, our approach focuses on matrix column-blocks and extends to both binary and ternary matrices. Additionally, their method is designed for matrix-to-matrix multiplication, whereas our method is for vector-to-matrix operations where the vector is a random $\mathbb{R}^n$ vector.

A related algorithm is the Mailman algorithm (Liberty & Zucker, 2009), which also reduces matrix-vector multiplication time by a logarithmic factor through preprocessing. However, its preprocessing differs from ours: the Mailman algorithm constructs a matrix $P$ by considering all possible binary-valued columns and expresses $A = U \times P$, enabling fast multiplication. In contrast, our approach constructs a list of permutations and segment sums, allowing us to achieve an $O(\log n - \log \log n)$ speedup even when multiplying a vector by the matrix $P$. Additionally, as discussed

in Section C.1, we design a segment matrix structurally similar to $P$ and leverage it in parallel settings.

## C. Proofs

**Theorem 3.6.** *Representing a binary or ternary weight matrix as block indices requires a space complexity $O\left(\frac{n^2}{\log(n)}\right)$, and the preprocessing algorithm to generate the block indices has a time complexity of $O(n^2)$.*

*Proof.* For Binary DNNs, let $B = W^\ell$ be a binary weight matrix. For Ternary DNNs, let a weight matrix $A = W^\ell$ be expressed as $A = B^{(1)} - B^{(2)}$ using Proposition 2.1. Then represent $A$ using the block indices of $B^{(1)}$ and $B^{(2)}$ produced by Algorithm 1. Let $B$ be either $B^{(1)}$ or $B^{(2)}$.

*Time Complexity:* During the preprocessing, we read all elements of the input matrix $B$, which requires $O(n^2)$ time. Blocking the matrix into $k$-Column Blocks has a time complexity linear to the number of blocks while determining the permutation corresponds to performing a bucket sort on the rows to achieve the lexicographic ordering, which requires $O(n)$. Consequently, the total preprocessing time is $O(n^2)$. Given that the input matrix size is $O(n^2)$, this complexity is asymptotically optimal, as any algorithm requires reading the matrix $B$ in $O(n^2)$.

*Space Complexity:* The output of the preprocessing phase is a permutation and a segmentation list as each block index. Recall that $|L_i| = |\mathcal{S}(B_i)| = 2^k$, where $k \leq \log_2(n)$. Therefore, storing each block index requires $O(n)$ space. The total number of column blocks is $\frac{n}{k}$. Hence, the total space complexity of the preprocessing output is $O(\frac{n^2}{k})$. Particularly, when $k = \log(n)$, the space complexity is $O(\frac{n^2}{\log(n)})$. As explained in Section 4, the inference algorithms only require access to the block indices. Therefore, replacing the weight matrices with the block indices, we *reduce the space complexity by a logarithmic factor.* $\square$

**Lemma 4.2.** $\vec{v} \cdot B_i = SS_{L_i, \sigma_{B_i}}(\vec{v}) \cdot Bin_{[k]}$.

*Proof.* Define $\vec{u} = SS_{L_i, \sigma_{B_i}}(\vec{v})$. At first, we should show that $|\vec{u}| = 2^k$, and as a result, the product on the right-hand side is valid. We know that $|SS_{L_i, \sigma_{B_i}}(\vec{v})| = |L_i|$ (see Definition 4.1). In addition, based on Definition 3.4, we know that in the Segmentation list, there is exactly one element for each possible binary value between 1 to $2^k$. So $|L_i| = |\mathcal{S}(B_i)| = 2^k \implies |\vec{u}| = 2^k$.

Now, let us focus on a single column of $B_i$ ($j$th column). We show that $\langle \vec{v} \cdot B_i[:, j] \rangle = \langle \vec{u} \cdot Bin_{[k]}[:, j] \rangle$, where $\langle \rangle$ is the dot product of two vectors. Based on the Definition 4.1,

we know that the $l$th element of $\vec{u}$ is the $l$th segment sum of $\pi_{\sigma_{B_i}}(\vec{v})$. This segment is the sum of some consecutive elements of $\pi_{\sigma_{B_i}}(\vec{v})$. Based on Definition 3.2, this permutation is designed such that similar rows of $B_i$ are positioned together. This means that all the elements of $\vec{v}$ that are in this segment (we denote this segment as $S_l$, in other words, $S_l$ is the indices of $\vec{v}$ that lie in this segment) are multiplied to the same binary value in $B_i$. Call this value as $b_l$. As a result, we can factor $b_l$ out for each segment of $\vec{v}$ in the dot product. We can rewrite the dot product as:

$$\langle \vec{v} \cdot B_i[:, j] \rangle = \sum_{q=1}^{n} \vec{v}[q] \cdot B_i[q, j] \tag{8}$$

$$= \sum_{l=1}^{2^k} \left( \sum_{e \in S_l} \vec{v}[e] \right) \cdot b_l = \sum_{l=1}^{2^k} \vec{u}[l] \cdot b_l$$

Where in the last line we used the fact that $\forall l, \vec{u}[l] = \sum_{e \in S_l} \vec{v}[l]$.

Each $b_l$ is the single that in column $j$ of $B_i$ corresponds to the segment $S_l$. Based on Definition 3.2, the applied permutation tries to sort all columns based on the lexicographic order of rows. So, this means that $Bin_{[k]}[l, j] = b_l$ and as a result,

$$\sum_{l=1}^{2^k} \vec{u}[l] \cdot b_l = \sum_{l=1}^{2^k} \vec{u}[l] \cdot Bin_{[k]}[l, j] \tag{9}$$

$$= \langle \vec{u} \cdot Bin_k[:, j] \rangle \tag{10}$$

As a result, for all columns $j$, the dot products are equal $\langle \vec{v} \cdot B_i[:, j] \rangle = \langle \vec{u} \cdot Bin_{[k]}[:, j] \rangle$. Consequently, the vector-matrix products are also equal. $\square$

### C.1. Parallelization

In this section, we discuss two approaches for parallelizing our algorithms to handle high-performance workloads.

**I. Parallelization based on block independence:** Our `RSR` and `RSR++` algorithms conduct each block computation in isolation, independent of the other block computations. Consequently, the block productions can be executed concurrently during the inference time. As a result, utilizing $c$ computation cores, the time complexities are further reduced by a $c$ factor, reducing the time complexities of `RSR` and `RSR++` to $O\left(\frac{n^2}{c(\log(n) - \log(\log(n)))}\right)$ and $O\left(\frac{n^2}{c \log(n)}\right)$, respectively. Additionally, further parallelization can be achieved by applying block production simultaneously on different columns of the binary-encoded matrix $Bin_{[k]}$.

---

**Algorithm 3** RSR++ (Step 2 in Inference Time)

---

1: **Input:** vector $\vec{u} \in \mathbb{R}^{2^k}$, and matrix $Bin_{[k]}$
2: **Output:** result $\vec{r} = \vec{u} \cdot Bin_{[k]}$
3: Initialize $\vec{x} \leftarrow \vec{u}$
4: **for** $i$ from $k$ to $1$ **do**
5:     $\vec{r}_i \leftarrow$ sum of odd indexed elements of $\vec{x}$
6:     Initialize $\vec{v}$ to a vector of size $\frac{|\vec{x}|}{2}$
7:     **for** $j$ from $1$ to $\frac{|\vec{x}|}{2}$ **do**
8:         $\vec{v}[j] \leftarrow \vec{x}[2i-1] + \vec{x}[2i]$
9:     **end for**
10: **end for**

---

**II. Parallelization on GPU:** Our second approach is based on the GPU implementation of our RSR algorithm discussed in Section 5.4 and Appendix E.3, and involves converting the segmentation and permutation steps during inference into a single 3D tensor multiplication. For each column-block $j$, we define a segmentation matrix $\mathcal{M}_j$ of size $n \times 2^k$, where each column is a one-hot vector of size $n$ indicating which element of the input vector $\vec{v}$ belongs to that segment (with a total of $2^k$ segments; see Section 4.1). Multiplying $\vec{v}$ by this matrix produces the vector $\vec{u}$ (see Algorithm 2 and Figure 1). The result is then multiplied by $Bin_{[k]}$. By stacking all $\mathcal{M}_j$ matrices across all $j$, we construct a 3D tensor $\mathcal{M}$. Precomputing $\mathcal{M} \times Bin_{[k]}$ allows us to reduce the inference process to a single tensor multiplication, significantly enhancing parallelization.

Future work could explore leveraging specific hardware instructions, such as blocking, permutations, segmentations, and block productions, to further optimize these steps at the hardware level in parallel.

## D. Discussion

In this section, we discuss some of the advantages and limitations of our proposed algorithms and outline potential directions for future research in this area.

### D.1. Advantages

- Our RSR++ algorithm shows up to a 29x improvement in inference time and a 6x reduction in memory usage, as observed in our experiments. Note that following our complexity analyses, these numbers get even larger by the size of the network layers. This significant improvement indicates that the algorithm can support larger hidden layers and matrices to enhance model accuracy while requiring fewer computational resources.
- The efficiency of our algorithm leads to reduced energy consumption, further contributing to the model's practicality and sustainability.
- Our approach enables the deployment of more advanced models on ordinary devices, such as per-

sonal computers, with limited memory and processing power.
- By processing one column block at a time, the algorithm only requires memory equivalent to the size of a single block, which is significantly less than that needed for full matrix multiplication.
- Last but not least, our method is deployable on top of any of the current or future binary or ternary models, without requiring any fine-tuning or re-training of the LLM or DNN. Once a model is trained, our preprocessing step can be applied once, allowing the preprocessed model to be utilized at any future point.

### D.2. Limitations

- PyTorch's hardware-level optimizations on GPUs make single vector-to-matrix multiplication extremely fast and efficient. The current implementation of our methods is at the application level and does not leverage hardware-specific instructions, which limits the speedup achieved when applied to real LLMs.
- Our approach demonstrates significant speedups when working with Python packages for matrices larger than $2^{10}$ in size. For smaller matrices, hidden constants and implementation overheads tend to dominate, reducing the practical benefits despite the theoretical guarantees. However, in real-world models, matrix sizes are typically much larger than this threshold, as discussed earlier.

## E. Experiments Details

### E.1. Synthetic Dataset

We built a synthetic dataset containing a set of short factual questions (ShortQuestions). In order to build this dataset, we used GPT-4 to give us a list of questions. For example, one sampled question from this dataset looks like: "What is the capital of France?", and the responses of both Standard and RSR models were "Paris". This is because we limited each model to generate only one token through a single feedforward pass.

### E.2. PyTorch Implementation Details: CPU Experiments

To further optimize Algorithm 2, for each column-block $j$, we constructed a segmentation matrix $\mathcal{M}_j$ of size $n \times 2^k$. Each column of $\mathcal{M}_j$ is a one-hot vector of size $n$, representing the indices of elements in $\vec{v}$ that belong to the corresponding segment before the permutation, with $2^k$ segments in total (see Figure 1). The inference process is then reduced to a matrix multiplication $\vec{v} \times \mathcal{N}$, followed by reshaping the output, where $\mathcal{N} = \mathcal{M} \times Bin_{[k]}$ and $\mathcal{M}$ is a

3D tensor containing all matrices $\mathcal{M}_j$ corresponding to the $\frac{n}{k}$ blocks. This approach takes advantage of the low-level optimizations already integrated into PyTorch. However, the baseline we compare against benefits from additional PyTorch-specific optimizations beyond this implementation.

The `forward()` function in the baseline model (`BitLinear`) uses a single vector-matrix multiplication in PyTorch:

```
1  def forward(self, input):
2      y = input @ self.ternary_matrix
3      return y
```

Listing 1: Simplified `forward` function in baseline

However, in the changed code, this function looks like this;

```
1  # self.rsr_matrix <= segmentation matrix
2  def forward(self, input):
3      y = (input @ self.rsr_matrix).permute
           (1, 0, 2).reshape(input.size(1),
           -1)
4      return y
```

Listing 2: Simplified `RSR` implementation

### E.3. PyTorch Implementation Details: GPU Experiments

To optimize the runtime of the previously discussed CPU implementation on the GPU, we parallelized the vector-matrix multiplication. Using the segmentation matrix $\mathcal{M}$, the multiplication was parallelized along its second dimension, corresponding to $k$ (see Section 4.2.2). To achieve this, we first expanded the input vector to a size $k$ times larger:

```
1      input_expanded = input.unsqueeze(1).
           expand(-1, k, -1)
```

Listing 3: `RSR` implementation on GPU (step 1)

Then, we ran a single multiplication of input_expanded and the matrix and finally reshaped the result:

```
1      expanded_result = torch.bmm(
           input_expanded, seg_matrix).squeeze
           (1)
2      result = torch.stack([expanded[:,j,:]
           for j in range(k)], dim=2).reshape(
           v.size(0), -1).unsqueeze(0)
```

Listing 4: `RSR` implementation on GPU (step 2)

This approach enables the computation of the product in parallel on the GPU. However, it does not utilize low-level optimized parallelization on the GPU, highlighting the need for future work on hardware-level optimizations for segmentation and permutation operations.

An evaluation of a single vector-matrix multiplication on GPU based on this implementations is provided in Appendix F.4.

## F. Extended Experiment Results

### F.1. Parameter $k$ Optimization

As outlined in Section 4.2.2, the optimal value of $k$ for `RSR` (reps. `RSR++`) is determined through a binary search over the interval $[0, \log(\frac{n}{\log(n)})]$ (resp. $[0, \log(n)]$). Once the optimal $k$ is identified for each $n$, it is then applied in the subsequent experiments. Figures 9a and 9b show the running time with respect to different $k$ values. By increasing $n$, the optimum value of $k$ also increases.

### F.2. `RSR` vs. `RSR++` on Native Implementation

In Section 5.1, we observed the performance gain of `RSR` and `RSR++` in comparison with the standard vector-matrix multiplication across different input sizes ($n$). In this section, we focus on the performance gain of `RSR++` versus `RSR` across different settings. To do so, we use our native `C++` implementation discussed in Section 5.1.

Figure 10 illustrates the difference between only `RSR++` and `RSR` for in the inference time. In this figure, we can see up to 25% improvement when using `RSR++` in the Step 2 of inference time (See Algorithm 3). The improvement percentage is computed using the formula $\frac{T(\text{RSR}) - T(\text{RSR++})}{T(\text{RSR})} \times 100$.

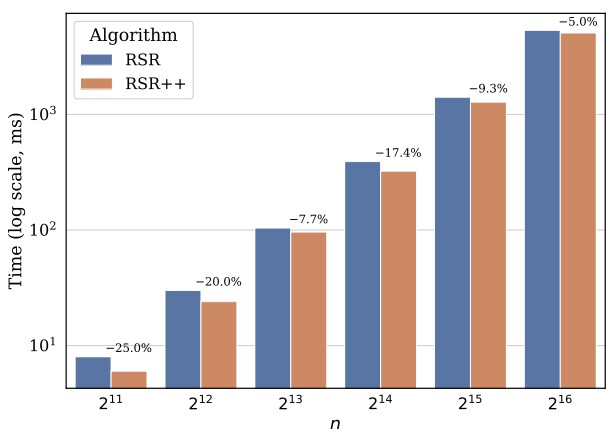

Figure 10: The comparison of `RSR` and `RSR++`. The percentage shows how much improvement we get using `RSR++` relative to `RSR`. This is a native `C++` implementation.

### F.3. Matrix Multiplication Using `NumPy`: Inference Time

In this section, we present the running time improvements achieved by `RSR`, on our implementation of `RSR` in Python

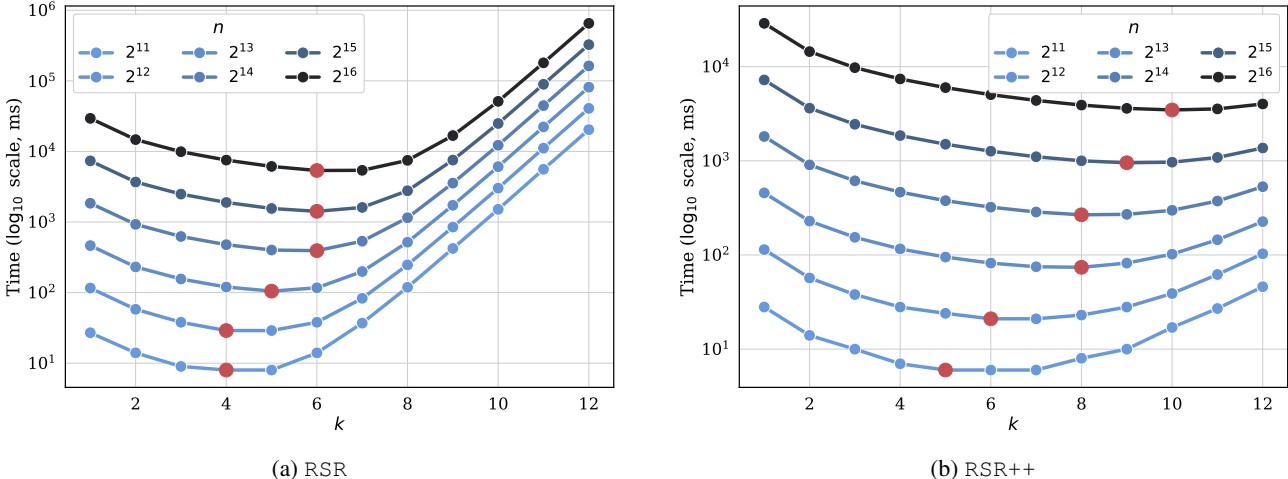

(a) RSR

(b) RSR++

Figure 9: Finding the optimum $k$ for each $n$. The red dots show the best $k$ for each $n$ that results in the best running time. Once calculating the optimum $k$ values, we use them for other experiments.

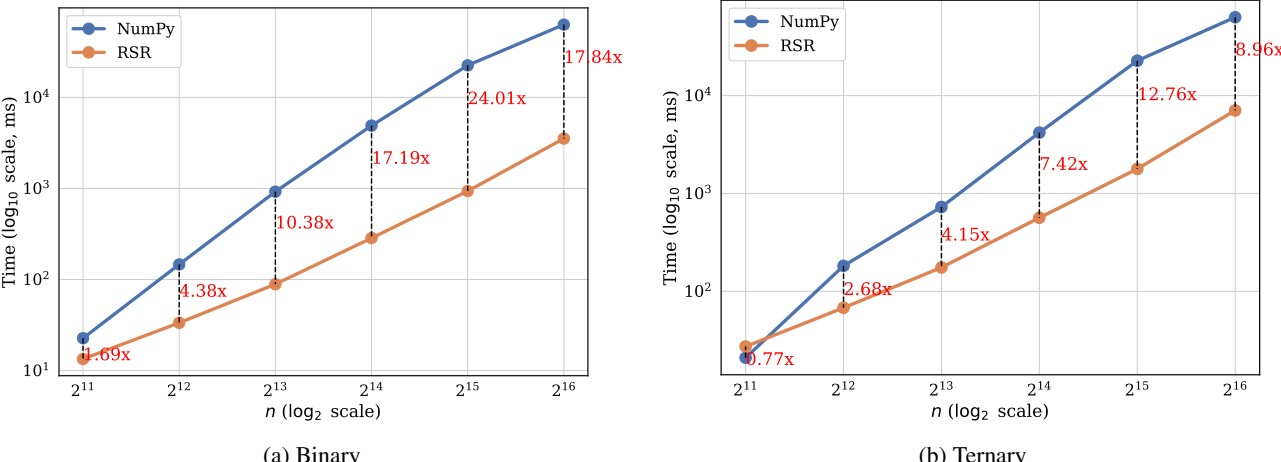

(a) Binary

(b) Ternary

Figure 11: Comparison of Time required to perform a vector to weight Matrix multiplication for (11a) Binary and (11b) Ternary weights between NumPy and RSR. Each data point is an average of 4 different runs.

using NumPy's built-in functionalities, discussed in Section 5.2.

As in previous sections, let $A^{n \times n}$ denote our weight matrix and $\vec{v}$ the input vector, where our goal is to compute $\vec{v} \cdot A$. We run the experiments on both Binary and Ternary matrices. This matrix is available prior to inference time, so we can run preprocessing on that. For the parameter $k$, we use the previously determined optimal values in Section F.1.

We initialize $n$ at $2^{11}$ and double its size in each experiment up to $2^{15}$. Each multiplication is executed four times, with results averaged to mitigate noise. Figures 11a and 11b present results for matrix multiplication using both the naive NumPy method and RSR for Binary and Ternary weight matrices, respectively. As shown, RSR achieves **up to a 24x**

**speedup** over the NumPy baseline on binary matrices of size $n = 2^{15}$, illustrating that our algorithm not only meets theoretical guarantees but also outperforms state-of-the-art practical methods.

### F.4. Performance evaluation on GPU

Using our GPU implementation in Appendix E.3, we evaluated the performance of a single vector-matrix multiplication on GPU. The results are provided in Figure 12. Here, we utilized matrices of size $2^{11}$ to $2^{14}$ and we can see up to **2x** speedup. However, as long as the $n$ (size of matrix) increases, the overhead of application-level optimization reducing the speedup compared to the optimized PyTorch standard implementation of vector-matrix product.

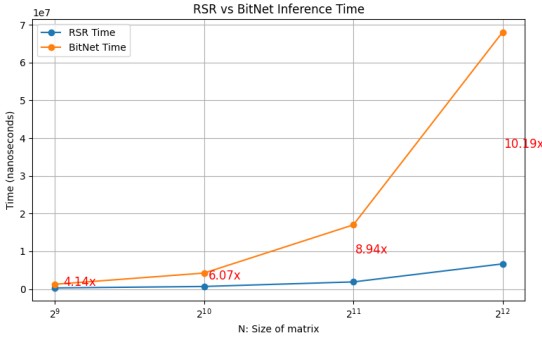

Figure 13: Time comparing RSR and original BitNet.cpp implementation.

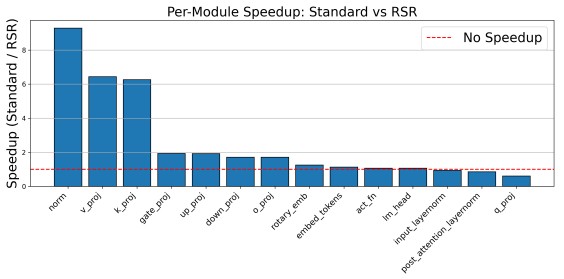

Figure 14: Per module speedup comparison of Standard and RSR augmented `Llama3-8B-1.58` model.

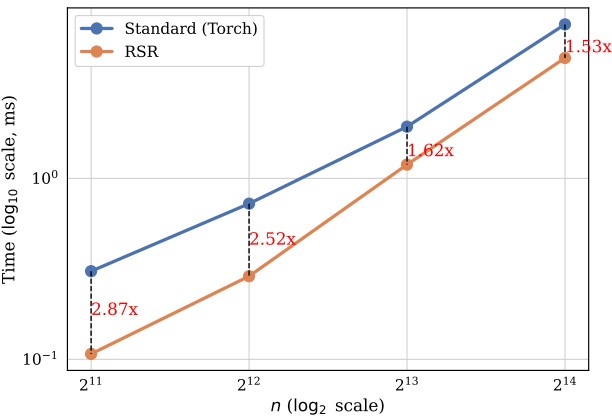

Figure 12: Comparison of time required to perform the vector-ternary-matrix multiplication on GPU.

## F.5. BitNet.cpp Comparison

In this section, we present an additional experiment to compare our ternary vector-matrix multiplication method with the optimized CPU implementation provided in BitNet.cpp (Wang et al., 2024a). The BitNet.cpp focuses primarily on optimizing the packing and unpacking of ternary weights. For instance, it uses the $I2_S$ kernel (Wang et al., 2024a; 2023), where every four 2-bit ternary values are packed into a single uint8 value (Wang et al., 2024a). Inference is then performed directly on these packed weights without unpacking, using an efficient custom kernel. We compare the performance of our matrix multiplication algorithm against this kernel across various matrix sizes. The results are shown in Figure 13.

Despite not implementing hardware-specific or low-level kernel optimizations in our implementation (RSR), we still observe a significant speedup in ternary matrix multiplication performance. We consider optimizing and utilizing kernel and hardware-specific instructions for RSR in future work, and we mainly focused on the algorithmic and application-level description in this work.

## F.6. Per Module Speedup

In this section, we analyze the speedup obtained for each individual PyTorch module in `Llama3-8B-1.58`. Figure 14 illustrates the speedup achieved by our method across different modules. The corresponding module names along with their roles in the Transformer architecture are summarized in Table 2.

| Module Name | Transformer Component |
|---|---|
| embed_tokens | Input Embedding Layer |
| rotary_emb | Rotary Positional Embedding (RoPE) |
| input_layernorm | LayerNorm before Attention |
| q_proj | Query Projection (Attention) |
| k_proj | Key Projection (Attention) |
| v_proj | Value Projection (Attention) |
| o_proj | Output Projection (Attention) |
| post_attention_layernorm | LayerNorm before MLP |
| gate_proj | Gated Linear Unit (GLU) Projection (MLP) |
| act_fn | Activation Function (MLP) |
| up_proj | Upward Projection (MLP) |
| down_proj | Downward Projection (MLP) |
| norm | Final LayerNorm (Post Decoder Blocks) |
| lm_head | Output Linear Projection to Vocabulary |

Table 2: Module names and their corresponding roles in the Transformer architecture.

