# OpenReview forum: "An Efficient Matrix Multiplication Algorithm for Accelerating Inference in Binary and Ternary Neural Networks"
_ICML.cc/2025/Conference — ICML 2025 poster_

### Official Review · Reviewer_czCs · 2025-03-13

**Overall Recommendation:** 3

**Summary:**

The paper proposes an efficient matrix multiplication algorithm for cases where one of the operands is binary or ternary. The binary weight matrices are partitioned into multiple blocks to exploit their intrinsic low-dimensional structure, leading to reduced storage and computation during inference. Within each partition, the rows are sorted lexicographically to form segments, enabling the storage of both the permutation and the segment row indices in the compressed binary matrix representation. During inference, the input is permuted and summed over each segment, with the exact output obtained by multiplying the binary matrix $Bin_{[k]}$. The proposed method achieves a memory and computation reduction of $\log n$. Compared to standard binary matrix multiplication, the algorithm delivers a 4-5x improvement in CPU latency for LLM inference.

**Claims And Evidence:**

The claims are supported by clear and convincing evidence.

**Essential References Not Discussed:**

The related works are addressed well.

**Experimental Designs Or Analyses:**

The experimental designs and analyses are generally sound and valid. However, the baseline implementation in the GPU runtime comparison section is somewhat unclear. I am curious whether the standard ternary matrix multiplication in Table 1 and Figure 12 was performed by casting the ternary weights to 16-bit or 32-bit floating point values.

**Methods And Evaluation Criteria:**

The proposed methods and evaluation criteria make sense. The RSR and RSR++ algorithms seem to improve efficiency. The runtime of RSR and RSR++ is reported in the paper.

**Other Comments Or Suggestions:**

Propositions 2.1 and 3.5, as well as Theorem 4.4, are missing proofs. While the proofs are straightforward, I believe each claim should be accompanied by its proof for completeness. Additionally, the proof of Theorem 4.3 appears before the theorem itself. It might be more appropriate to place the theorem at the beginning of Section 4.2.1, followed by its proof.

**Other Strengths And Weaknesses:**

**Strengths**

- The paper is clearly written and easy to follow, with examples that effectively illustrate the concept of the proposed method.
- The proposed method demonstrates a meaningful reduction in CPU latency for LLM inference, which could have a substantial impact on running LLMs efficiently on edge devices such as laptops or mobile phones.

**Weaknesses**

- RSR and RSR++ involve permutation of the input data, which may result in additional memory I/O overhead.
- It is unclear whether the proposed method is applicable to tiled matrix multiplication, a common GEMM implementation for multicore CPUs or GPUs.

**Questions For Authors:**

1. Was the standard ternary matrix multiplication in PyTorch performed by casting the ternary weights to 16-bit or 32-bit floating point, or was a custom kernel used for ternary weights?
2. In the last sentence of Section 5.3, it is stated that the size of the segmentation matrix is identical to that of the ternary weight matrix in the PyTorch implementation. Could the authors provide more clarification on why this is the case?
3. Does the algorithm require any modifications to be adapted for tiled GEMM?

**Relation To Broader Scientific Literature:**

The proposed method could potentially be applied to graph processing or graph-related applications, such as Graph Neural Networks [Zhou et al., 2020] and Low-Density Parity Check (LDPC) decoding [Gallager, 1962].

[Zhou et al., 2020] Zhou, Jie, et al. "Graph neural networks: A review of methods and applications." *AI Open*, 2020.

[Gallager, 1962] Gallager, Robert. "Low-density parity-check codes." *IRE Transactions on Information Theory* 8.1 (1962): 21-28.

**Theoretical Claims:**

I checked the correctness of Theorem 3.6 and Lemma 4.2 and they seem correct. However, the proof of Theorem 4.4 is missing and I think it should be formally introduced.

---

> ### Author Rebuttal · Authors · 2025-03-28
>
> **Theoretical Claims:**
>
> Thanks for bringing this to our attention. We explicitly include all the proofs in the updated version. The proof of Theorem 4.4 was removed due to its similarity to Theorem 4.3 (with a different value of $k$).
>
> **Weaknesses:**
>
> *RSR and RSR++ involve permutation of the input data...*
>
> Thank you for raising this important concern regarding the memory overhead introduced by permutations. We would like to clarify that after applying RSR during the preprocessing phase, the inference stage does not require storing the full-weight matrix. Instead, we store only the segmentation and permutation information. This design leads to a logarithmic ($log n$) reduction in memory usage during inference, as illustrated in Figure 5. (for the parallel approach we store a new matrix $\mathcal{M}$ as mentioned in the answer to Reviewer #2).
>
> Moreover, in an optimized implementation, we avoid explicitly permuting the input vector. Rather than generating a permuted copy, we perform **index-based lookups** to access the necessary vector elements on the fly. This avoids both memory overhead and unnecessary data movement.
>
> As described in Section 4.1 (immediately before Section 4.2), we introduce an **in-place vector lookup** strategy that enables access to the indices of a permutation without reconstructing the permuted vector.
>
> **Other Comments or Suggestions:**
>
> Thanks for the suggestions. We will change the organization of the proofs accordingly.
>
> **Questions For Authors:**
>
> *1:*
>
> We kindly refer you to our response to Reviewer #1’s last suggestion for additional context on this point.
>
> In standard PyTorch implementations for 1.58-bit LLMs (e.g., using formats like I2_S [1][2]), the weights are stored in a packed format—typically encoding four 2-bit values into a single uint8. However, during inference, these packed weights are unpacked and converted into bfloat16 format for computation (**nn.BitLinear** module).
>
> In contrast, the more efficient bitnet.cpp implementation—which we included in our updated experiments (see the response to Reviewer #1’s suggestion)—performs matrix multiplication directly on the packed uint8 weights, avoiding the unpacking step. Even in this optimized setting, our RSR algorithm shows a significant speedup in matrix multiplication (a PoC of RSR is implemented alongside bitnet.cpp; please refer to the anonymous repository). This is evident in the inference time comparison plot available at:
> [https://anonymous.4open.science/r/RSR-1566/bitnetcpp/figures/inference_time_plot.png](https://anonymous.4open.science/r/RSR-1566/bitnetcpp/figures/inference_time_plot.png)
>
> *2:*
>
> Thank you for pointing this out, and we apologize for the lack of clarity. The reference was intended to describe the segmentation matrix $\mathcal{M}$, which is introduced to support parallelism, as discussed in Appendix C.1. To enable compatibility with underlying parallelism, we transform the original matrix $M$ into a tensor $\mathcal{M}$, reducing the vector-matrix multiplication to a vector-tensor multiplication (see also our response to Reviewer #2).
>
> We will revise the main text to explicitly define the segmentation matrix and clarify its role. Importantly, $\mathcal{M}$ occupies the same amount of memory as the original matrix $M$, since it contains the same number of elements—only its shape differs.
> In more detail, the tensor $\mathcal{M}$ is generated by stacking $n / k$ smaller matrices of shape $n * 2^k$ (Appendix C.1). The size of this tensor is: $n * 2^k * n / k$. After multiplying this 3D tensor to the $Bin_{[k]}$ matrix (which has the size $2^k * k$), we create a matrix of size: $n * n/k * k = n^2$ which is stored in the preprocessing phase and used in the inference as a replacement for original matrix $M$.
>
> *3:*
>
> Please also refer to our final response to Reviewer #2 for a discussion on how RSR supports parallelization across different phases of the algorithm.
>
> As described in Appendix C.1, during the preprocessing phase—where we have full access to the weight matrices—we construct a segmentation matrix $\mathcal{M}$. This tensor is then stored and used for inference. During inference, any parallelization strategies that applies to standard vector-matrix multiplication (e.g., tiling in GEMM) remain fully applicable. That is, the same tiling mechanisms can now be applied to $\mathcal{M}$ instead of the original matrix $M$.
>
> Importantly, $\mathcal{M}$ has the same total size as the original matrix $M$, but a different shape. In fact, for the experiments in Section 5.4 (LLM on GPU), the PyTorch’s default parallelization mechanisms are already being applied on top of the RSR-transformed matrices.
>
> *References:*
>
> * [1] Wang, Jinheng, et al. "1-bit AI Infra: Part 1.1, Fast and Lossless BitNet b1. 58 Inference on CPUs." arXiv preprint arXiv:2410.16144 (2024).
> * [2] Ma, Shuming, et al. "The era of 1-bit llms: All large language models are in 1.58 bits." arXiv preprint arXiv:2402.17764 1 (2024).

---

> > ### Comment · Reviewer_czCs · 2025-04-03
> >
> > Thank you for the thorough response to my questions. All of my concerns are addressed, and I am inclined to keep my score.

---

### Official Review · Reviewer_xCtm · 2025-03-14

**Overall Recommendation:** 4

**Summary:**

This paper describes and complexity-analyzes an efficient algorithm (RSR/RSR++) for multiplication with binary (and by derivation, ternary) matrices. CPU and GPU implementations are provided, and inference with ternary-weight LLMs is demonstrated.

**Claims And Evidence:**

Yes.

**Essential References Not Discussed:**

I believe the relevant prior studies on efficient binary matrix multiplication are discussed.

**Experimental Designs Or Analyses:**

Yes, they are fine.

**Methods And Evaluation Criteria:**

Yes.

**Other Comments Or Suggestions:**

N/A

**Other Strengths And Weaknesses:**

N/A

**Questions For Authors:**

N/A

**Relation To Broader Scientific Literature:**

Though there exist efficient binary matrix multiplication algorithms, this study presents thorough complexity analysis and practical implementations, as well as real-world ternary LLM inference experiments.

**Theoretical Claims:**

I can follow the general logic but might not have spotted minor mistakes.

---

> ### Author Rebuttal · Authors · 2025-03-28
>
> Thank you for your time and the careful review of our work. We appreciate it.

---

> > ### Comment · Reviewer_xCtm · 2025-04-07
> >
> > I thank the authors for the rebuttal, and I maintain my original evaluation.

---

### Official Review · Reviewer_APww · 2025-03-14

**Overall Recommendation:** 4

**Summary:**

This work proposes an efficient algorithm for binary/ternary matrix multiplication, where the binary/ternary weights are known ahead of time. The algorithm first transforms the ternary matrix into a sum of binary matrices, and then compresses the binary matrices into a set of aggregate values which contribute to an entry in the result vector. This is done by segmenting the matrix into blocks, ordering the rows lexically, and storing counts for the number of occurrences of binary values across different rows. They also design an efficient multiplication algorithm that leverages this compressed format, which can achieve O(n^2/log(n)) runtime with respect to the matrix dimensions. This leads to practical speedups of >5x for matrix-vector multiplication.

## Update after rebuttal

The authors addressed my concerns, and I am inclined to keep my score (accept).

**Claims And Evidence:**

Yes - the measured runtime for the method clearly demonstrate the practical speedups which are attainable through their approach.

**Essential References Not Discussed:**

I am not aware of any missing essential references.

**Experimental Designs Or Analyses:**

The CPU benchmarking setup and GPU benchmarking setup both seem reasonable.

**Methods And Evaluation Criteria:**

Yes - the included benchmarking is reasonable for both end-to-end latency and matrix-vector latency measurements (included for both GPU and CPU). For example, they included both native C++ benchmarking and numpy benchmarking (showing that they compare with high-performance kernels), as well as GPU-level benchmarking (at the application level, without custom kernel implementations).

**Other Comments Or Suggestions:**

N/A

**Other Strengths And Weaknesses:**

Strengths:
- The proposed algorithm attains complexity improvements relative to prior multiplication algorithms for binary/ternary weights.
- The algorithm is fully parallelizable and can therefore be mapped to both CPU and GPU implementations.
- The paper explains their method both through algorithms as well as clear visualizations.

Weaknesses:
- While the GPU implementation attains speedups while implemented only at the application level, a kernel-level implementation would be necessary for harnessing greater speedups from the algorithm.

**Questions For Authors:**

- In Section 5.3, the performance was evaluated using 3 datasets. However, unless I am missing something, the accuracy should be identical with and without this algorithm (aside from minor numerical differences). Is there a reason for evaluating using all 3 datasets?
- The proposed algorithm requires the full matrix to be compressed together - is it therefore more challenging to employ particular tensor parallelism strategies which split along the column dimension (i.e. which allocate different rows to different GPUs)?

**Relation To Broader Scientific Literature:**

The paper presents a fast algorithm for matrix-vector multiplication with binary/ternary weights. For the majority of binary/ternary quantization works, the weights are the matrix compressed in binary/ternary, and this would be available ahead of time, making this algorithm applicable. This work can therefore be used as a drop-in approach to accelerate inference with these techniques.

**Theoretical Claims:**

I went through the outlined methodology and algorithms and did not identify any issues.

---

> ### Author Rebuttal · Authors · 2025-03-28
>
> **Weaknesses:**
>
> *While the GPU implementation attains speedups...*
>
> Thank you for the insightful suggestion—this is indeed a valid point. In this work, our primary focus was on the algorithmic design and application-level evaluation of the RSR algorithm. We compared its performance across various settings using existing, standard matrix multiplication implementations in C++, NumPy, and PyTorch.
>
> Following Reviewer #1’s recommendation (Reviewer HdLU), we also included a kernel-level comparison between RSR and bitnet.cpp, which is specifically optimized for ternary matrix multiplication at this lower level [1]. Even in this setting, RSR demonstrates a notable speedup, highlighting its potential for further performance gains.
>
> The code for this new experiment is available in the updated anonymous repository, which contains a low-level implementation of RSR as a PoC ([https://anonymous.4open.science/r/RSR-1566/bitnetcpp/README.md](https://anonymous.4open.science/r/RSR-1566/bitnetcpp/README.md)).
>
> This observation suggests that RSR could benefit from additional system-level enhancements. As part of future work, we can explore low-level system optimizations, including custom kernel implementations and hardware-specific instructions tailored to RSR’s structure.
>
> **Questions For Authors:**
>
> *In Section 5.3, the performance was evaluated using 3 datasets...*
>
> You are absolutely right—the performance should match the standard setting exactly, as our proposed matrix multiplication algorithm is an exact method for computing the product. The inclusion of multiple datasets is intended to show the generalizability of our approach across different scenarios and model architectures. It also emphasizes that the implementation is independent of the specific dataset, with only minor variations in speedup due to noise across different implementations.
>
> *The proposed algorithm requires the full matrix to be compressed together...*
>
> Thank you for highlighting this important point. We agree that parallelism plays a key role in standard neural network implementations. In our work, we addressed this in Section C.1 of the Appendix, where we discuss how our algorithm supports parallelization.
>
> To clarify, we assume full access to the weight matrices during the **preprocessing phase.** In this phase, we transform the original matrix $M$ into a new form by constructing a segmentation matrix $\mathcal{M}$, then, in the inference, we just calculate the multiplication of vector to $\mathcal{M}$. Any system-level parallelism (as discussed in Section C.1), is still applicable to the multiplication of the vector to the new tensor, similar to the standard multiplication. This transformation is for the case when we want to leverage from underlying parallelism. Otherwise, our standard inference procedure—as described in Algorithm 2—uses permutations and segmentation lists, which is not necessarily compatible with underlying parallelism (if any).
>
> If there is existing parallelism in the underlying libraries (e.g., row- or column-wise parallelism applied during inference), our approach remains compatible. Since $\mathcal{M}$ retains the same size as $M$, just with a different shape, the existing parallelism mechanisms can still be applied effectively. We can observe this fact in Figure 12 and Table 1, where the GPU parallelization exists, and we still have a notable speedup.
>
> The transformation (from $M$ to $\mathcal{M}$) happens once during preprocessing, when the full matrix is assumed to be available, while parallelism is typically applied at inference time, distributing matrix tiles across available GPU cores. Our method does not interfere with this process and remains fully compatible with standard parallel inference execution.
>
> If there is a parallelism strategy applied in the preprocessing time (when storing and preprocessing the weights, i.e. not in the inference time), then our approach should be applied in any individual blocks (either row or column blocks, depending on the parallelism). Then, in the inference time, any individual block multiplication can be translated to RSR multiplication.
>
> *References:*
>
> * [1] Wang, Jinheng, et al. "1-bit AI Infra: Part 1.1, Fast and Lossless BitNet b1. 58 Inference on CPUs." arXiv preprint arXiv:2410.16144 (2024).

---

### Official Review · Reviewer_HdLU · 2025-03-14

**Overall Recommendation:** 3

**Summary:**

The paper introduces two algorithms, RSR and RSR++, designed for accelerating matrix multiplication in binary/ternary LLMs during inference. By preprocessing weight matrices into column blocks, permuting rows lexicographically, and computing segmented sums to create optimized indices, the proposed methods reduce time complexity and memory usage by a logarithmic factor. Experiments have been conducted on both cpus and gpus, and demonstrate up to 29x faster inference, 6x memory reduction, and 5.24x speedup in 1.58-bit LLM inference.

## update after rebuttal
I thank the authors for the thorough rebuttal. My concerns regarding the implementation have been largely addressed (e.g., comparison with BitNet.cpp). However, I still do not understand why the authors only evaluated their method on the curated ShortQuestions dataset instead of providing results using a more commonly used evaluation dataset for the rebuttal. Therefore, I maintain my original rating of weak accept.

**Claims And Evidence:**

Yes, most claims made in the submission are supported by clear and convincing evidence.

**Essential References Not Discussed:**

NA

**Experimental Designs Or Analyses:**

Yes. See "Methods And Evaluation Criteria" for issues.

**Methods And Evaluation Criteria:**

Most of them are standard that make sense. However, I am wondering why the authors only considered comparing the cases of limiting
each model to generate only one token through a single feedforward pass and even created the ShortQuestions synthetic dataset for this purpose. The authors should also perform experiments that generate multiple tokens to present the impacts of the proposed algorithms during the auto-regressive generation stage of LLMs that include the usage of KV cache, which also has vector-matrix multiplication workloads. This stage is actually the main bottleneck in LLM inference. Moreover, for the case of only generating one token (which I assume it refers to the prefilling stage), the latency will only be a bottleneck in real-world applications when the context is very long, which is more similar to matrix-matrix multiplication instead of vector-matrix multiplication. The authors should explain their experimental designs in detail, performing the suggested additional experiments or providing a convincing explanation on why they are not performed.

**Other Comments Or Suggestions:**

Please provide breakdowns of latency comparison for different LLM components, including attentions, FFNs, and others.

**Other Strengths And Weaknesses:**

- The preprocessing and segmented sum approach effectively exploits the fixed structure of quantized weights. The logarithmic-factor improvements over naive multiplication are impressive.
- A detailed theoretical complexity analysis, supported by proofs, supports the claimed efficiency gains.

**Questions For Authors:**

How is the baseline 1.58-bit model implemented for LLM inference on CPU (Figure 6 results)? Please provide more details on the baseline CPU implementation. Specifically, the official implementation, bitnet.cpp (https://github.com/microsoft/BitNet), has already applied some optimizations for CPU inference. Is bitnet.cpp already applied in the "torch" implementation described in the paper? If not, please provide additional experimental results that compare with the bitnet.ccp baseline as well.

**Relation To Broader Scientific Literature:**

The work addresses a critical challenge (LLM inference efficiency) with a theoretically sound and empirically validated solution, which has high potential impact for deploying quantized LLMs on resource-constrained devices.

**Theoretical Claims:**

I went over the theoretical claims and did not find noticeable issues.

---

> ### Author Rebuttal · Authors · 2025-03-28
>
> **Methods And Evaluation Criteria:**
>
> Thank you for the helpful suggestion. We agree that performance evaluation in other settings—such as generating full sequences rather than single tokens—would also provide valuable insights. We will include additional experiments in the updated version to show the performance and speedup in individual components of the LLM (such as KV Cache, attention, etc).
>
> Regarding implementation, we currently replace all **nn.Linear** modules with our proposed RSR-based implementation. This module is the building block of any Fully Connected component in an LLM model, which contains all the vector-matrix multiplications applied during inference. We will clarify this detail in the updated Supplemental Material.
>
> As a result, all the components, such as KV cache computations and attention mechanisms, are already running under the new algorithm in the reported statistics. As suggested, in the updated version, we will profile these components **individually** and report performance metrics and speedups for each (rather than just the end-to-end performance comparison).
>
> **Questions For Authors:**
>
> *How is the baseline 1.58-bit model implemented...*
>
> That's a good question. For the baseline 1.58-bit model inference on CPU (Figure 6), we followed the standard implementation provided in [2], as well as the [official blog post](https://huggingface.co/blog/1_58_llm_extreme_quantization) by the authors of [2]. Specifically, we used the same inference process described in those sources. To ensure the model runs on the CPU, we explicitly configured PyTorch to use the CPU backend by disabling the CUDA environment. For the GPU experiments (Figure 12), we allowed PyTorch to utilize CUDA as expected. We will clarify these implementation details in the updated version.
>
> *Additional experiment on BitNet.cpp implementation...*
>
> Thank you for pointing this out. We have conducted an *additional* experiment to compare our ternary vector-matrix multiplication method with the optimized CPU implementation provided in [bitnet.cpp](https://github.com/microsoft/BitNet). The bitnet.cpp implementation focuses primarily on optimizing the **packing and unpacking** of ternary weights. For instance, it uses the I2_S kernel [1], where every four 2-bit ternary values are packed into a single uint8 value [1]. Inference is then performed directly on these packed weights without unpacking, using an efficient custom kernel.
>
> We have compared the performance of our matrix multiplication algorithm against this kernel across various matrix sizes. The results are shown in the following plot:
> [https://anonymous.4open.science/r/RSR-1566/bitnetcpp/figures/inference_time_plot.png](https://anonymous.4open.science/r/RSR-1566/bitnetcpp/figures/inference_time_plot.png)
>
> These results are also documented in our updated anonymous repository:
> [https://anonymous.4open.science/r/RSR-1566/bitnetcpp/README.md](https://anonymous.4open.science/r/RSR-1566/bitnetcpp/README.md)
>
> Despite not implementing hardware-specific or low-level kernel optimizations in our implementation (RSR), we still observe a significant speedup in ternary matrix multiplication performance. We consider optimizing and utilizing kernel and hardware-specific instructions for RSR in the future work and we mainly focused on the algorithmic and application-level description in this work.
>
> Regarding the standard PyTorch-based implementation described in the original paper ([2]), we clarify that this implementation uses packed ternary weights for storage. However, during inference, the packed uint8 values are unpacked and converted into bfloat16, and the multiplication is performed in the unpacked domain. This applies to the experiments shown in Figures 11 and 12.
>
> We will include this clarification and the bitnet.cpp comparison in the updated manuscript. Thank you again for your valuable suggestion.
>
> *References:*
>
> * [1] Wang, Jinheng, et al. "1-bit AI Infra: Part 1.1, Fast and Lossless BitNet b1. 58 Inference on CPUs." arXiv preprint arXiv:2410.16144 (2024).
>
> * [2] Ma, Shuming, et al. "The era of 1-bit llms: All large language models are in 1.58 bits." arXiv preprint arXiv:2402.17764 1 (2024).

---

### Decision · Program_Chairs · 2025-05-01

**Decision:**

Accept (poster)

**Comment:**

All reviewers advocated acceptance of the work, with two weak accepts and two accepts following the rebuttal. Reviewers consistently found the work well written and explained, including with diagrams. Reviewers generally found the empirical results and evaluation metrics presented as convincing as to practical speedups being demonstrable with the proposed methodology, on both CPU and GPU. In fact better results would be expected with low-level implementation as discussed by the reviewers and authors. Reviewers did have some concerns including the practical usage with KV cache for multiple tokens, the lack of clarity on GPU baselines, and comparisons to other implementation, but these appear to have been addressed by the authors. With the topic of LLM inference acceleration being of great interest in the research community, the clearly written paper, and the extensive analysis and results I believe this paper is a solid contribution to ICML.